# In Situ Experimental Study on the Active Support Used for Building Rectification

**DOI:** 10.3390/ma13092015

**Published:** 2020-04-25

**Authors:** Krzysztof Gromysz

**Affiliations:** Department of Building Structures, Faculty of Civil Engineering, Silesian University of Technology, Akademicka 5, 44-100 Gliwice, Poland; krzysztof.gromysz@polsl.pl; Tel.: +48-32-237-11-27

**Keywords:** building rectification, deflected building, stiffness, support

## Abstract

There are many vertically deflected building structures in the world that require rectification. Temporary supports installed in the building bearing walls can be used to perform such a rectification. The supports consist of a hydraulic piston jack, a stack of parallelepiped steel elements, and a concrete grout. The structure is unevenly raised and reaches the desired vertical position using such supports. The support in which the piston extension is forced at the given time is an active support. The aim was to determine the stiffness of an active support. The investigations were performed in in situ conditions during experimental building rectification. No such investigations have been performed to date. It has been demonstrated that the stiffness of the investigated support results from the stiffness of the serially connected elements forming the support. In general, the support stiffness depends on the value of the force occurring in the support and is rising linearly along with the load for the investigated range. It was also shown that the force existing in the active support also depends on the stiffness of the building being rectified. The investigations carried out show that it is advantageous to use supports with smaller stiffness for rectification, as forces with smaller values must be induced in them. The application of forces with lower values also allows the avoidance of unfavorable penetration of the unlifted part of the building into the ground.

## 1. Introduction

One of the issues emerging in civil engineering is the vertical deflection of building structures. The issue concerns all types of buildings. For example, [1] shows vertically deflected residential buildings; a case of a vertically deflected silo was analyzed in [2], and a vertically deflected neo-Gothic church was analyzed in [3]. The most renown deflected structures are towers. These include, for example, the Leaning Tower of Pisa [4] and a historical bell tower in Silesia [5]. Most often, the deflection is caused by uneven ground subsidence. For example, uneven subsidence caused by the soil pressure relief in the excavation area was analyzed in [6]. However, uneven subsidence is caused most often by the presence of soil with insufficient bearing capacity. A building on collapsible loess is shown for example in [7]. For structures placed on soils with insufficient bearing capacity, see [8]. Uneven subsidence can cause the failure of building elements, as shown in [9].

The deflection of adjacent buildings may lead to their interaction. Such a situation took place as a result of simultaneous changes of deformations and slope of the ground surface [10]. The deflection may also be caused by the deformation of the above-ground part of the structure caused by earthquakes. The above-ground part of the structure is then deformed, as a result of which the adjacent structures can be subject to pounding. This phenomenon was analyzed for buildings in series [11], steel buildings [12] and multi-story buildings in series considering the effect of infill panels [13]. This issue also relates to superstructure segments in a multi-supported elevated bridge [14] and typical concrete box-girder bridges [15]. To avoid damages caused by pounding, a sufficient distance is provided between adjacent buildings taking into account their structural solutions [16] and in its absence, in special situations, the structures are moved [10]. Numerical analyses of deflections of the buildings located in close proximity to each other were carried out, among others, in [17]. On the other hand, an analysis of building deflection between floors using a discrete five-floor building model is presented in [18]. The impact of underground mining is a frequent cause of building deflection, where uneven land subsidence is caused by the tightening of voids left after underground exploitation, and even the exploitation of natural underground water bodies [19]. The impact of underground coal mining on the deflection of buildings [20,21], changes in land sloping near flood areas [22] and land subsidence as a result of groundwater overpumping were forecast in the investigations [23].

Tilted structures are most often stabilized to prevent the deterioration of deformations. Such method was applied for the Leaning Tower of Pisa [24] where the subsoil was stabilized, and for historical sites at Auschwitz Birkenau [25], where the position of the walls was stabilized. The rectification of the individual building wall [26] or entire buildings [27] is carried out in special cases. Soil is then removed from underneath the higher part of the structure [28] or the lower part of the structure is lifted. 

Such experimental removal of tilting by lifting was carried out in Silesia, Poland. In this case, hydraulic piston jacks were installed in the load-bearing walls of the building cellar. The building was detached with their help (Figure 1a). The horizontal detachment ran through all the elements of the building and divided the building into two parts: the unevenly lifted part and the part remaining in the ground. Displacements were applied to the individual pistons of the jacks during such uneven lifting. The building was lifted at the average height of 177 mm and the maximum height was 354 mm (Figure 1b). As the extension of the jack pistons was limited to 200 mm, it was necessary to install a stack of steel parallelepiped elements under some of them. The stack and the jacks were placed on a sheet of metal embedded in a concrete grout. Therefore, the temporary support of the building consisted of concrete grout with an embedded sheet metal, a jack, and possibly a stack of parallelepiped steel elements.

The pistons of the jacks were extended while removing the deflection. The support, in which the piston extension (*u*_ext_) was forced in the given cycle, was called an active support (Figure 2). There was an increase in the value of the force (*Q*) in this support. Moreover, in the place where the active support was installed, there was an upward movement of the elevated part of the building (*u*_obj_) and a slight downward movement of the part remaining in the ground (*u*_fou_). Moreover, as the support force was rising, the support was shortened (Δ*l*_sup_). The support in which no piston extension was forced, was called a passive support. In the process of uneven lifting, depending on the lifting phase, the given support can be either active or passive.

The active support and the parts of the building in the place where this support is installed are the subject of the investigations outlined in this article. The aim of the investigations is to define the model of the support and of the part of the building in the place of its installation and to determine the parameters of the support and building model. Until now, studies on vertical displacement have been limited in civil engineering to structural elements, with regard to the lift-slab technology [29,30]. The issue of stability of the reinforced concrete columns being compressed was mainly addressed in the investigations [31,32], where the effect of elements forming the stack on its bearing capacity was examined. Moreover, the investigations of eccentrically loaded walls with the filled joints [33] and of masonry structures with dry contacts points are known [34]. The studies of the stiffness of a stack of steel elements forming part of the support have only been presented in [35] and in [36] for wooden stacks in laboratory conditions. The investigations of supports installed in buildings in situ conditions were not carried out. The knowledge of the stiffness of these supports is essential for the safe design and implementation of the building rectification process.

## 2. Research Program

The building where the investigations were planned and carried out is completely cellared and has five above-ground floors. Its height above the ground level is 16.1 m (Figure 1). The projection of the building is represented by a rectangle with the sides of 10.42 m and 12.05 m (Figure 3). The load-bearing cellar walls are concrete and are 0.4 m and 0.35 m thick. The load-bearing walls of the above-ground floors are made of solid brick and aerated concrete blocks and are 0.38 m thick. The ceilings between the floors are Ackerman-type closely ribbed. The strip foundations are executed as reinforced concrete foundations. The section of those foundation under the external walls is rectangular with the dimensions (*b*/*h*) 1.0 m/0.4 m, and under the internal walls, with the dimensions 1.4 m/0.4 m. The subsoil layers below the foundations were determined with geotechnical boreholes drilled to the depth of 6 m. There is a 0.5 m thick layer of sand with a degree of density of *I*_D_ = 0.7 directly underneath the strip foundations. Below, sandy clay in a hard-plastic state with a plasticity degree of *I*_L_ = 0.2 is deposited to the depth being recognized.

During the investigations, the building rested on 36 temporary supports, whose location is shown in Figure 4. The distance between them was from 1.29 m to 2.63 m. The supports consisted of a hydraulic piston jack and a 10 mm thick sheet metal embedded in a 150 mm thick concrete grout. In addition, parts of the supports consisted of stacks of parallelepiped steel elements with the dimensions of 350/320/72.5 mm made of rolled profiles. The base of the supports was at the level of −2.40 m (Figure 5).

The measurements of displacements of the building elements in the place where the support no. 29 was installed and of changes in the length of the elements of this support were designed. This was the place where the structure was raised by the highest value—354 mm. The following measurements were carried out (Figure 5): forced stroke of the jack piston (*u*_ext_), forces occurring in the support (*Q*), displacement of the part of the building remaining in the ground (*u*_fou_), displacement of the straightened part of the building in relation to the part of the building remaining in the ground (*u*_o-f_), change of the length of the temporary support (Δ*l*_sup_), change of the length of the jack and the stack of steel parallelepiped elements (Δ*l*_jack_ + Δ*l*_und_). The measurement of displacement *u*_fou_ was designed in relation to a steel rod driven into the ground at a depth of 1.5 m (Figure 5). It was assumed that during the investigations, the rod is not displaced and is a fixed point. Other displacements and changes in length were measured in relation to the building structure’s elements and in relation to elements of the supports.

It was assumed that the measurements will be carried out during the monotonic increase of the force in the support no. 29. The value *Q* increased from *Q*_min_ to *Q*_max_. The value *Q*_max_ was constant and was 780 kN. On the other hand, *Q*_min_ assumed the values of 100 kN, 200 kN, 300 kN, and 400 kN. 

## 3. Progress and Results of Measurements

The oil pressure in the jack increased by means of a hydraulic system used during the previous building rectification. Piston extension was forced this way and the force *Q* increased. After the force in the support no. 29 was induced with the maximum value (*Q*_max_), it was reduced to the minimum value (*Q*_min_). For technical reasons, force reduction was carried out in two ways. First, the force in the adjacent support no. 31 (Figure 4) increased, as a result of which the force in the tested support has decreased. The support no. 29 was then becoming a passive support. After a force of 680 kN was applied in the adjacent support, oil was drained from the jack of the support no. 29 until the force *Q*_min_ was reached in that support. The implemented program of force inducement in the support 29 is shown in Figure 6a where the time is shown on the horizontal axis and the load value *Q* on the vertical axis.

The following load cycles (*Q*_min_–*Q*_max_) are distinguished in this system: 100–780 kN, 200–780 kN, 300–780 kN, and 400–780 kN. The load value increased at an average speed of 5.8 kN/s. The measured values were recorded at a frequency of 10 Hz.

The values of forces during the investigations were measured using strain gauges, which were located between the jacks and the elevated part of the building. The measuring range of the jack was 700 kN and the indication accuracy was 1%.

Piston extension, displacements, and length changes were measured using LVDT transducers. The measurement error, resulting from the non-linearity of the transducer, was up to 0.5%. The building resting on the jacks at the time of measurement is shown in Figure 7a, and support no. 29 during the investigations is shown in Figure 7b.

Assuming that the positive return of the vertical axis is directed upwards (Figure 6b), the displacements *u*_obj_ in the place of installation of the active support assume positive values and the displacements *u*_fou_ assume negative values. An increase in the value *Q* is accompanied by a decrease in the support length, so Δ*l*_sup_ takes negative values for the active support. The value of vertical displacement of the elevated part of the building is
(1)$$u_{\mathrm{obj}}=u_{\mathrm{ext}}-\left|u_{\mathrm{fou}}\right|-\left|\Delta l_{\sup}\right|$$

The difference in the displacements *u*_obj_ and *u*_fou_ is represented by the displacement of the elevated part of the building in relation to the part remaining in the ground and is designated by *u*_o-f_.
(2)$$u_{o-f}=u_{\mathrm{obj}}-u_{\mathrm{fou}}$$

On the other hand, the change in the support length equals the sum of changes in the length of its constituent elements
(3)$$\Delta l_{\sup}=\Delta l_{\mathrm{jack}}+\Delta l_{\mathrm{und}}+\Delta l_{\mathrm{grout}}$$
where:Δ*l*_jack_—change in the jack length,Δ*l*_und_—change in the length of the stack of parallelepiped elements,Δ*l*_grout_—change in the length of the concrete grout.


### 3.1. Displacements of Building Elements

Figure 8a shows a change in the value of the force *Q* as a function of the forced piston extension *u*_ext_. The bold line distinguishes fragments of the dependency *Q*−*u*_ext_ corresponding to a monotonically increasing load from *Q*_min_ to *Q*_max_. It can be seen that the piston extension, when decreasing the load, is significantly influenced by the way the force *Q* in the support 29 is reduced. First, when the decrease in the force value in the support 29 was the effect of increasing the force in the adjacent support 31, the piston of the jack in the support 29 was further extended. Then, when the value of the force in the jack was reduced by draining the oil from it, the piston extension was reduced.

Figure 8b shows the analogous dependency *Q−u*_o-f_. It distinguishes the values corresponding to a monotonically increasing load from the value *Q*_min_ to *Q*_max_. In this case, it can also be seen that the way the load is reduced has a significant effect on the values of displacements *u*_o-f_. When the decrease in the force value in the support 29 was the effect of increasing the force in the jack of the support 31, the value *u*_o-f_ continued to increase. When the value of the force in the jack was reduced by draining the oil from it, the value *u*_o-f_ was reduced. When referring the changes in the value *u*_o-f_ to the changes in the value *u*_o-f_ shown in Figure 8a, one can see that the changes of displacements *u*_o-f_ are smaller than the corresponding changes *u*_ext_.

In turn, Figure 8c shows the dependency *Q*−*u*_obj_, whereas the values *u*_obj_ are determined from the dependency (1). When comparing the dependency *Q−u*_obj_ with the dependency *Q−u*_ext_, one can see that only part of the piston extension shows itself as an upward movement of the object.

The dependencies *Q*−*u*_fou_ were provided in Figure 8d. It can be seen that the increase in the value of the force *Q* is accompanied by the settlement of the foundation under the jack. In the beginning, when increasing the load, the displacements *u*_fou_ are small. Then, a further increase in the value *Q* is accompanied by a more significant increase in the displacements *u*_fou_. The remark applies to each load cycle *Q*_min_–*Q*_max_.

Table 1 lists the values of extension *u*_ext_ and displacements *u*_o-f_, *u*_obj_, *u*_fou_ corresponding to the change Δ*Q* in the range of *Q*_min_–*Q*_max_. The stiffness *k* of the system is then defined as
(4)$$k=\frac{\Delta Q}{\Delta u_{\mathrm{ext}}}$$
where a Δ*u*_ext_ corresponds to the change of Δ*Q*.

The stiffness *k* of the system is influenced by the properties of all its constituent elements: the part of the building being lifted, the part remaining in the ground, the ground and the properties of the supports, in particular the investigated active support. 

The following was additionally defined: *k*_obj_—stiffness of the elevated part of the building
(5)$$k_{\mathrm{obj}}=\frac{\Delta Q}{\Delta u_{\mathrm{obj}}}$$

*k*_fou_—stiffness of the part of the building remaining in the ground
(6)$$k_{\mathrm{fou}}=\frac{\Delta Q}{\Delta u_{\mathrm{fou}}}$$
and stiffness
(7)$$k_{o-f}=\frac{\Delta Q}{\Delta u_{o-f}}$$

The stiffnesses described by the dependencies (5)–(7) were determined for the four examined load cycles *Q*_max_*−Q*_min_ and listed in Table 1. 

The stiffnesses are graphically presented in the diagrams in Figure 9. The values of the parameters defined by the dependencies (5)–(7) are not constant and depend on the force amplitude (Δ*Q* = *Q*_max_ − *Q*_min_). With the increase of the value *Q*_min_ (with constant *Q*_max_ = 680 kN), higher values of the parameter were obtained. The smallest values, from 99 MN/m to 154 MN/m, are assumed by *k*. On the other hand, the highest values, from 895 MN/m to 2223 MN/m, are assumed by *k*_fou_. The stiffness *k*_obj_ assumes the values from 169 MN/m to 250 MN/m and *k*_o-f_ from 142 MN/m to 225 MN/m. The greatest variability, together with the load, is characteristic for *k*_fou_.

The influence of the value of the force *Q* on the system parameters was analyzed. For this purpose, the displacement values corresponding to the force *Q* of 100 kN, 200 kN, 300 kN, 400 kN, 500 kN, and 600 kN were read from a load curve of 100 kN–680 kN. Displacement increments (Δ*u*_ext_, Δ*u*_o-f_, Δ*u*_obj_, Δ*u*_fou_), corresponding to increments Δ*Q* of the force, were determined on such basis for the ranges of 100–200 kN, 200–300 kN, 300–400 kN, 400–500 kN, and 500–600 kN. For these ranges, the stiffnesses *k*, *k*_f-o_, *k*_obj_ and *k*_fou_ were determined according to the dependency (4)–(7). The determined displacement and stiffness values are listed in Table 2 and presented graphically in Figure 10.

It can be seen that the stiffness *k*_fou_ is most dependent on the load value. Its value decreases significantly as the load increases, falling from 32,452 MN/m (for *Q* from 100 kN to 200 kN) to 646 MN/m (for *Q* from 500 kN to 600 kN). Other parameters of the structure do not show such significant changes along with the change *Q*. It can be concluded that all stiffnesses, except for *k*_fou_, slightly grow with an increase in the value *Q* from 100 kN to 200 kN and then stabilize. The stiffness *k* rises from 73 MN/m to 115 MN/m, *k*_o-f_ from 109 kN/m to 153 kN/m and *k*_obj_ from 110 kN/m to 198 kN/m.

### 3.2. Change in the Length of the Support

Figure 11a shows a change in the length of the support (Δ*l*_sup_) caused by a change in the value *Q*. A load increasing monotonically from the value *Q*_min_ to *Q*_max_ was distinguished there. The aim of the investigations is to analyze variations in the length of the support and its elements corresponding to changes in force in the range of *Q*_min_–*Q*_max_.

Figure 11b shows the measured change in the length of the jack and a stack of four parallelepiped steel elements arranged on each other (Δ*l*_sup_ + Δ*l*_jack_) caused by a change in the force *Q*. An increase in the load *Q* resulted in shortening the length of the elements. It can be noticed that this shortening is not fully proportional to the increase in the value of the force *Q*. This is because a change in the stack length is caused by three deformations. The first one is the deformations of the material of parallelepiped elements. The second one is the mutual displacement of the profiles, from which a parallelepiped element is created by welding. The third one is the deformations occurring around interfaces of parallelepiped elements which are not perfectly contiguous. The latter are characterized by a non-linear force-deformation dependency.

Figure 11c shows the change of the jack length (Δ*l*_jack_) under the influence of the increasing load *Q*. The change Δ*l*_jack_, obviously, is smaller than the changes shown in Figure 11b. The measured change of the jack length, corresponding to a change in the force *Q* in the range of 100 kN to 780 kN, is 0.912 mm. Such a significant change in the jack length derives from the fact that it is the result of both the deformations of the cylinder material as well as the deformations occurring at the contact between the sheet metal fixed in the concrete grout and the jack base. In turn, Figure 11d shows a diagram describing the shortening of the concrete grout (Δ*l*_grout_) under the influence of the load. The value (Δ*l*_grout_) was not the subject of direct measurement, but it was determined from the dependency (3).

Table 3 lists changes in the length of the support and its elements corresponding to changes in the range of *Q*_min_–*Q*_max_. The stiffnesses of the support were then defined
(8)$$k_{\sup}=\frac{\Delta Q}{\Delta l_{\sup}}$$
and of the support elements
(9)$$k_{\mathrm{jack}}=\frac{\Delta Q}{\Delta l_{\mathrm{jack}}},\;k_{\mathrm{und}}=\frac{\Delta Q}{\Delta l_{\mathrm{und}}},\;k_{\mathrm{grout}}=\frac{\Delta Q}{\Delta l_{\mathrm{grout}}}$$
where *k*_jack_ is the stiffness of the jack, *k*_und_ stiffness of the stack of parallelepiped elements, *k*_grout_ stiffness of the grout.

The stiffnesses of the support and its elements determined according to (8) and (9) were compiled in Table 3 and presented graphically in diagrams from Figure 12. The values of the parameters defined by the dependencies (8) and (9) are not constant and depend on the force amplitude (Δ*Q* = *Q*_max_ − *Q*_min_). With the increase of the value *Q*_min_, with constant *Q*_max_ equal to 680 kN, higher values of the parameters were obtained. The smallest values, from 307 MN/m to 457 MN/m, are assumed by *k*_sup_. On the other hand, the highest values are assumed by the stiffness *k*_grout_ and *k*_und_; from 1541 MN/m to 1700 MN/m and from 916 MN/m to 1832 MN/m, respectively. The stiffness *k*_jack_ assumes the value from 658 MN/m to 1020 MN/m.

The influence of the value of the force *Q* on the support parameters was then analyzed. For this purpose, the displacement values corresponding to the force *Q* of 100 kN, 200 kN, 300 kN, 400 kN, 500 kN, and 600 kN were read from a load curve of 100–600 kN. Displacement increments (Δ*l*_sup_, Δ*l*_jack_, Δ*l*_und_, Δ*l*_grout_), corresponding to increments Δ*Q* of the force, were determined on such basis for the ranges of 100–200 kN, 200–300 kN, 300–400 kN, 400–500 kN, and 500–600 kN. For these ranges, the stiffnesses *k*_sup_, *k*_jack_, *k*_und_ and *k*_grout_ were determined according to the dependency (8) and (9). The determined displacement and stiffness values are listed in Table 4. The values are presented graphically in Figure 13. 

The stiffnesses corresponding to the centers of the analyzed ranges were marked with points. The figures clearly show that the stiffness *k*_sup_ of the support increases with the increase in the force *Q*. This stiffness increases from 213 MN/m with the load *Q* of 100–200 kN to 430 MN/m with the load *Q* of 500–600 kN. The greatest change in stiffness is distinct for the stack of steel elements, whose stiffness increases from 443 MN/m for *Q* in the range of 100–200 kN to 1391 MN/m for *Q* in the range of 500–600 MN/m. On the other hand, the force *Q* has the smallest effect on the jack stiffness, which varies from 503 MN/m for *Q* for the range of 100–200 kN to 985 MN/m for *Q* for the range of 500–600 kN.

## 4. Analysis of the Results of the Investigations

### 4.1. Dependencies between the Analyzed Stiffnesses

Figure 14a summarizes the stiffness of the system *(k*), building structure elements (*k*_fou_, *k*_obj_) and supports (*k*_sup_) corresponding to the ranges Δ*Q*. The relationship between these stiffnesses results from Equations (4)–(6) and (8) and from the fact that the displacement *u*_ext_, corresponding to the stiffness of the whole system, is equal to the sum of displacements of the part remaining in the ground (*u*_fou_), the lifted part (*u*_obj_) and the change in the length of the support (Δ*l*_sup_)—Equation (1). Hence, the stiffness *k* results from the serial combination of the stiffness *k*_fou_*, k*_obj_, and *k*_sup_, thus
(10)$$k=\frac{k_{\mathrm{obj}}k_{\mathrm{fou}}k_{\sup}}{k_{\mathrm{obj}}k_{\mathrm{fou}}+k_{\mathrm{obj}}k_{\sup}+k_{\mathrm{fou}}k_{\sup}}$$

The stiffness *k* can also be expressed by the stiffness *k*_o-f_
(11)$$k=\frac{k_{o-f}k_{\mathrm{jack}}}{k_{o-f}+k_{\mathrm{jack}}}$$
where *k*_obj_, *k*_fou_, *k*_sup_, *k*_o-f_ are dependent on *Q*. The values *k*, calculated according to (4), (10) and (11), are compiled in Table 5 and presented graphically in Figure 15. The values *k*, determined according to these dependencies, do not differ significantly, which proves the correctness of the adopted model. At the same time, it should be noted that the stiffness *k* of the model increases from 73 MN/m with the force *Q* between 100 kN and 200 kN and then stabilizes for the value *Q* greater than 200 kN, assuming the values between 104 MN/m and 115 MN/m. This is mainly influenced by the stiffness of the lifted part, which at *Q* of about 200 kN stabilizes with a value of about 200 MN/m. The stiffness of the support (*k*_sup_) consists of the stiffness of the jack (*k*_jack_), a stack of elements installed under the jack (*k*_und_) and grout (*k*_grou*t*_). The values of these stiffnesses, corresponding to the examined ranges Δ*Q*, are shown in Figure 14b.

The elements forming the support rest on top of each other, hence, according to (3), the total change in the length of a support is the sum of changes in the length of its constituent elements. Thus, the support stiffness can be determined from the binding dependency of the three connected stiffnesses in series
(12)$$k_{\sup}=\frac{k_{\mathrm{jack}}k_{\mathrm{und}}k_{\mathrm{grout}}}{k_{\mathrm{jack}}k_{\mathrm{und}}+k_{\mathrm{jack}}k_{\mathrm{grout}}+k_{\mathrm{und}}k_{\mathrm{und}}}$$
where the stiffnesses *k*_jack_, *k*_und_, and *k*_grout_ are dependent on the load *Q*. The model of the system adopted above is shown in Figure 16.

### 4.2. Influence of Stiffness of the Constituent Elements on Model Stiffness 

The influence of the particular parameters on the system stiffness (*k*) was analyzed based on the model defined in Figure 16. The influence of a stack of steel elements and concrete grout as well as the of the stiffnesses *k*_fou_ and *k*_obj_ were studied successively.

The dependency *Q*−*k*, determined based on (10), was provided in black in Figure 17a, assuming that the support is made up of a jack placed on the grout (no parallelepiped elements; *k*_und_ = ∞). In addition, the line to which the values refer (support with four parallelepiped elements) is marked in grey. The dependencies obtained are similar in nature. It is shown by comparing the respective values that the removal of the parallelepiped elements has increased the system stiffness by 13%–55%.

The analogous diagram is given in Figure 17b, by assuming that the support does not include a grout (*k*_grout_ = ∞ black line). It can be stated, by referring the values obtained in this way to the results obtained for the support with the grout (grey line), that the removal of the grout has increased the stiffness of the system by 5%–10%.

However, assuming that the supports rest on an undeformable substrate (*k*_fou_ = ∞; Figure 17c)*,* the stiffness of the system increases by 10%–35% depending on the force *Q*.

The greatest increase in the system stiffness—by about 300%, was achieved assuming that the stiffness *k*_obj_ assumes an infinitely high value. The above assumption is tantamount to assuming that both the part of the building being raised is a rigid body and all the adjacent supports are infinitely rigid elements.

The stack in the investigations consisted of four parallelepiped elements, 72.5 mm long each. Hence, the stack length was 290 mm. The results obtained can be generalized into stacks consisting of a different number of elements. Table 6 compiles the calculated stiffnesses *k*_und_ corresponding to the stacks the part of which is *n* = 1, …, 10 elements, considering that the elements of the stack are connected in series. The stiffnesses *k*_und_, as load-dependent values, are shown in Figure 18a. In turn, Table 7 and Figure 18b show the stiffnesses *k*_sup_ calculated according to (12). The case for *n* = 0 corresponds to the situation where the support consists only of grout and a jack. Within the investigated range, the stiffness of the support may vary from 124 MN/m (*n* = 10, Δ*Q* = 100–200 kN) to 985 MN/m (*n* = 0, Δ*Q* = 500–600 kN).

## 5. Conclusions

One of the issues in civil engineering is the vertical deflection of building structures. For buildings, such deflections can be removed by means of temporary supports installed into its walls. Temporary supports consist of hydraulic piston jacks, stacks of steel parallelepiped elements and a concrete grout. With their help, the building is torn apart in a horizontal plane. Then, the part of the building situated under the tearing area is elevated unevenly until a vertical position is reached. The support in which the piston extension is forced at the given stage is an active support.

Piston extension in the support increases the force value in this support. The investigations performed in situ conditions reveal that the force increase in the support is unfavorable, as it results in the deformation of the building part being lifted and in the unfavorable movement of the part remaining in the ground towards the bottom. The force increase in the active support depends on the stiffness of the system consisting of the unevenly elevated part of the building, the part remaining in the ground and temporary supports. The stiffness of the whole system in the place and direction of piston extension of the active support for the range of 100–200 kN was 73 MN/m and for the load range of 500–600 kN it was 115 MN/m.

The stiffness of the support itself has the greatest influence on the system stiffness in the place and direction of piston extension of the active support. The stiffness of the support consisting of a jack, a stack of four parallelepiped elements with a total length of 290 mm and 150 mm thick grout depends on the load value. For the load range of 100–200 kN it was 213 MN/m and for the load range 500–600 kN it was 430 MN/m.

The support stiffness relies upon the stiffness of the serially connected elements forming the support. Concrete grout has the greatest stiffness. For the load range of 100–200 kN, its stiffness was 2258 MN/m and for the load range 500–600 kN, it was 1692 MN/m. The jack with the stiffness of, respectively, 503 MN/m and 985 MN/m, had lowest stiffness. On the other hand, the stiffness of the stack consisting of four parallelepiped elements was 443 MN/m for the load range of 100–200 kN and 1391 MN/m for the load range of 500–600 kN.

The defined model of the support and the stiffnesses of its elements determined during the in situ tests made it possible to calculate the stiffness of the supports which are also characterized by other lengths of the stack of steel elements. The support stiffness is declining along with the growing length of the stack of parallelepiped elements.

Apart from the active support, the stiffness of the elevated building part resting on supports has the greatest influence on the stiffness of the entire system. This was 110 MN/m for the load range of 100–200 kN and 194 MN/m for the load range of 500–600 kN. In the case where the stiffness of this part reaches an unlimited value, then the stiffness of the system would increase three times. This would be the case if the stiffness of the adjacent passive supports was unlimited.

There have been no studies up to now of buildings rectified by uneven lifting. Therefore, the results provided in this work also allow conclusions to be drawn about the rectification process. When using supports with smaller stiffness, it is necessary to induce forces with smaller values, which is beneficial. However, the use of supports with insufficient stiffness can be dangerous due to the loss of system stability. Moreover, to ensure that the unlifted part of the building does not deepen into the ground, it is necessary to change the values of forces in the supports by as little as possible. This means that small extensions have to be applied to the jack pistons due to the fact that while the forces in the supports are growing, the stiffness of the part of the building remaining in the ground is decreasing.

## Figures and Tables

**Figure 1 materials-13-02015-f001:**
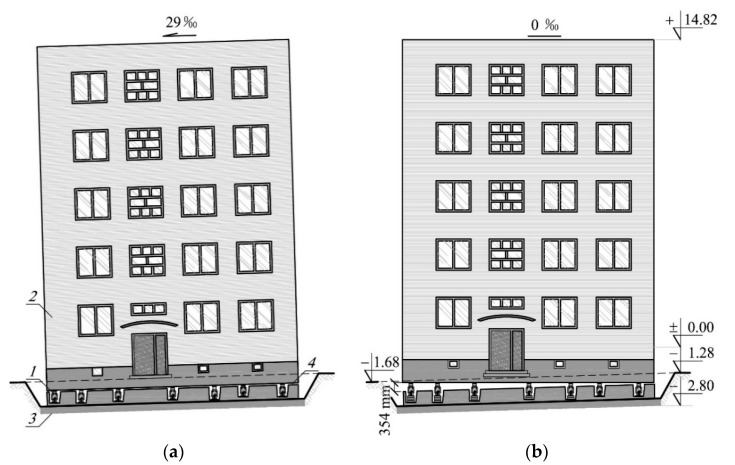
Façade of the building the tilt of which was removed: (**a**) jacks installed in the building walls, (**b**) the building after uneven lifting; *1*—temporary support, *2*—unevenly lifted part of the building, *3*—part of the building remaining in the ground, *4*—detachment plane of the building.

**Figure 2 materials-13-02015-f002:**
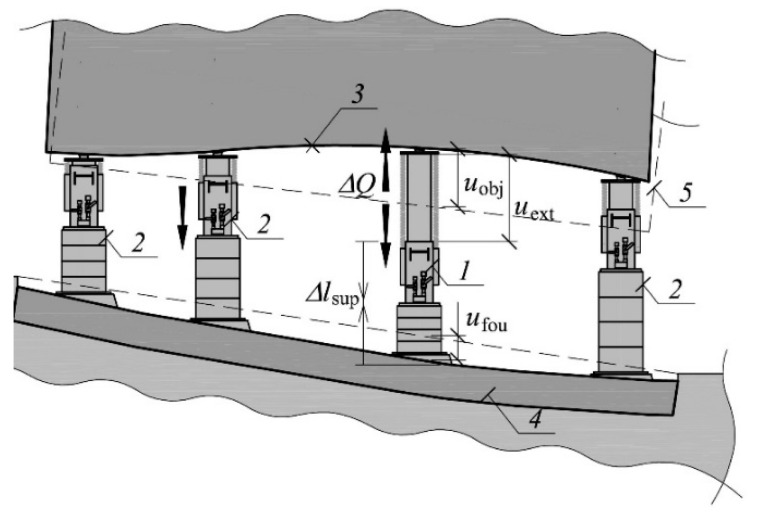
Change in the length of the active support and displacements of building elements in the place of its installation: *1*—active support, *2*—passive support, *3*—unevenly lifted part of the building, *4*—part of the building remaining in the ground, *5*—original position of the part of the building, *u*_ext_—forced extension of the active support piston, *u*_obj_—displacement of the lifted part of the building, *u*_fou_—displacement of the unlifted part, Δ*l*_sup_—change in the length of the support, Δ*Q*—change in the value of force in the support.

**Figure 3 materials-13-02015-f003:**
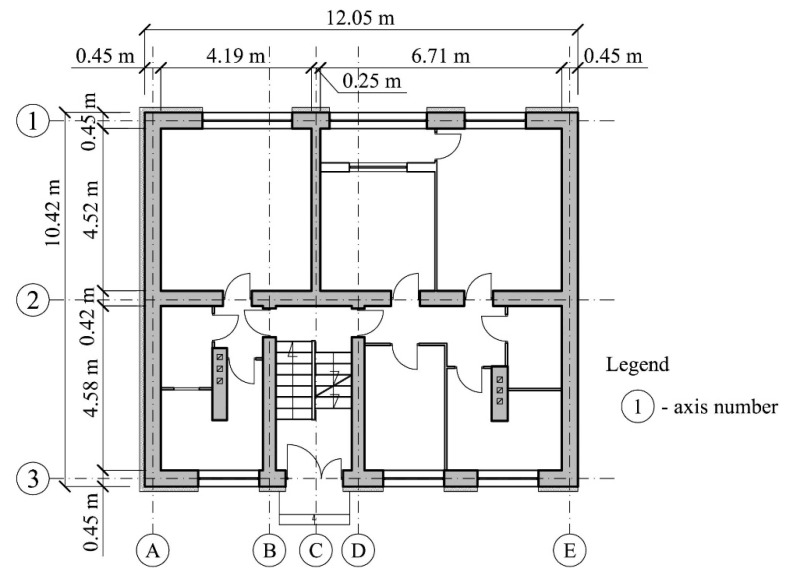
Ground floor plan.

**Figure 4 materials-13-02015-f004:**
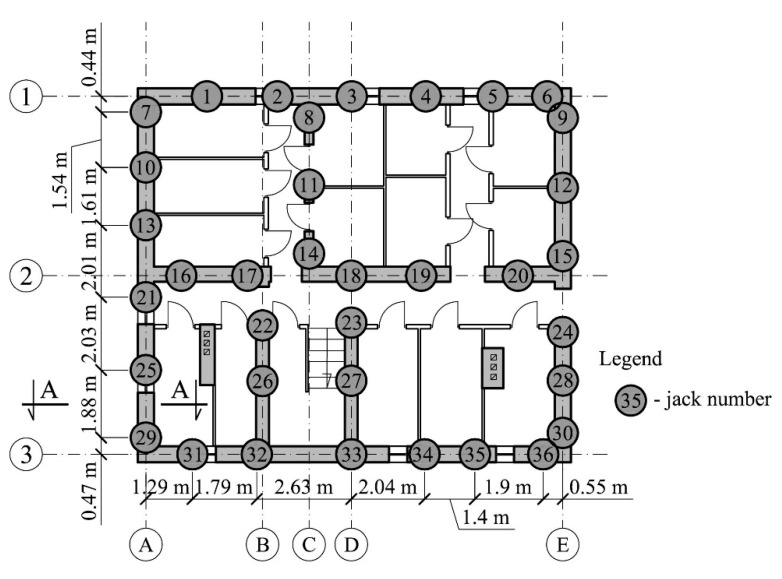
Arrangement of jacks installed in the cellar walls (section A—A is shown in Figure 5).

**Figure 5 materials-13-02015-f005:**
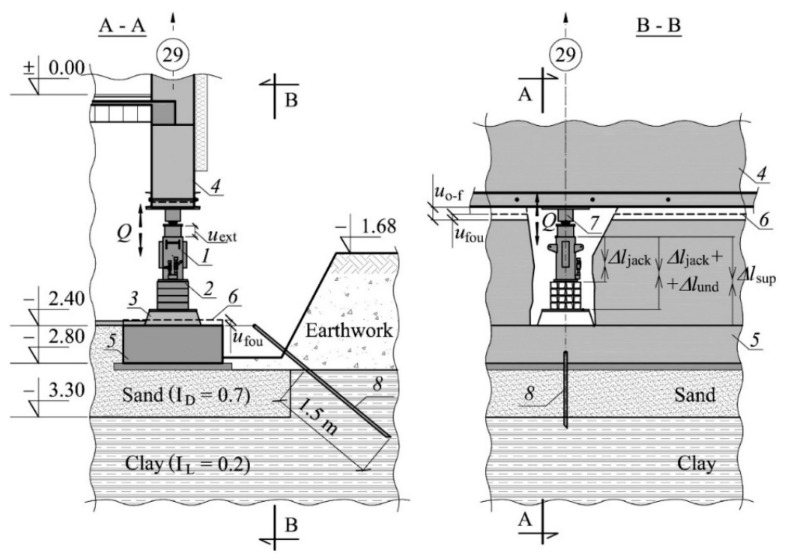
The following measurements were carried out: the forced jack piston extension (*u*_ext_), displacement of building elements (*u*_o-f_*, u*_fou_), changes in support length (Δ*l*_sup_), changes in length of support elements (Δ*l*_jack_ + Δ*l*_und_, Δ*l*_jack_) and force in the support (*Q*): *1*—jack, *2*—stack of parallelepiped elements, *3*—concrete grout, *4*—elevated part of the building, *5*—part of the building remaining in the ground, *6*—original position of the part of the building, *7*—dynamometer, *8*—steel rod driven into the ground.

**Figure 6 materials-13-02015-f006:**
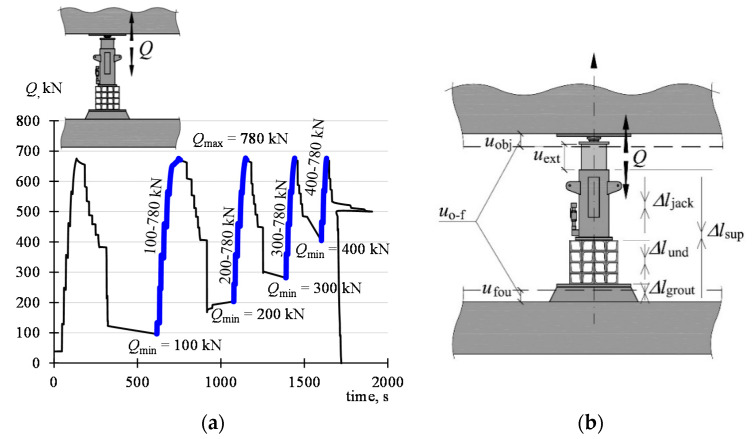
Test program: (**a**) cycles of the change of the value *Q* of the force in support no. 29 (phases of the monotonic load increase being the subject of analysis are in bold), (**b**) symbols of movements of the building parts and changes in the length of support elements.

**Figure 7 materials-13-02015-f007:**
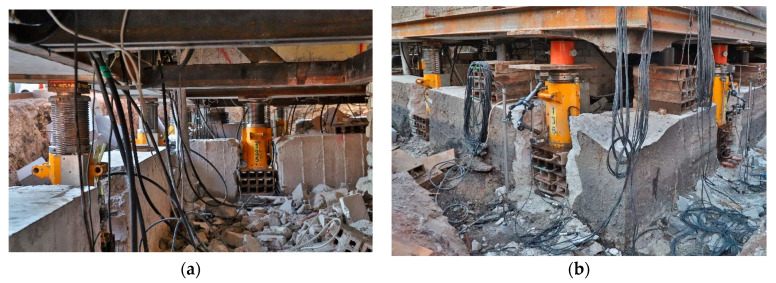
Building during measurements: (**a**) unevenly lifted part resting on the jacks, (**b**) support no. 29 during measurements.

**Figure 8 materials-13-02015-f008:**
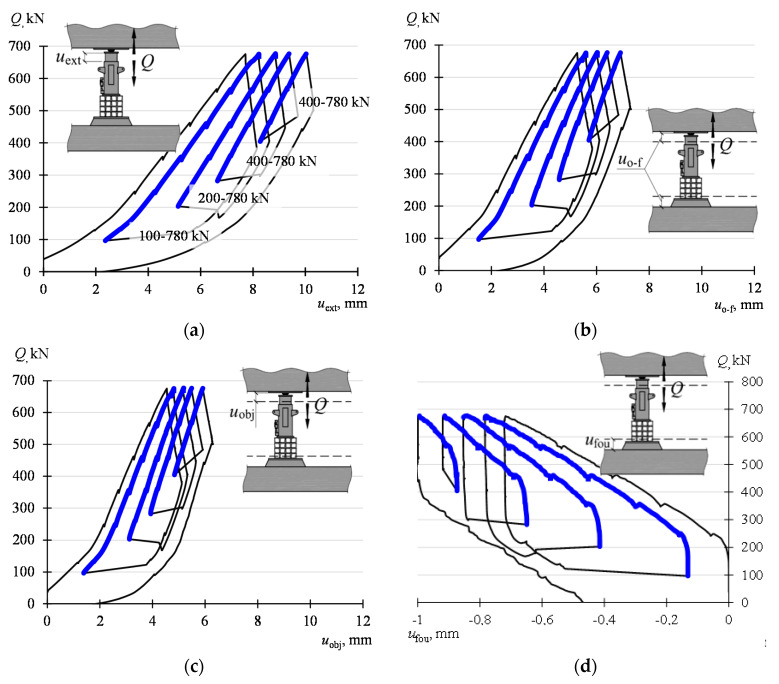
Force—extension and force—displacement dependencies of the building structure elements: (**a**) *Q*−*u*_ext_, (**b**) *Q*−*u*_o-f_, (**c**) *Q*−*u*_obj_, (**d**) *Q*−*u*_fou_.

**Figure 9 materials-13-02015-f009:**
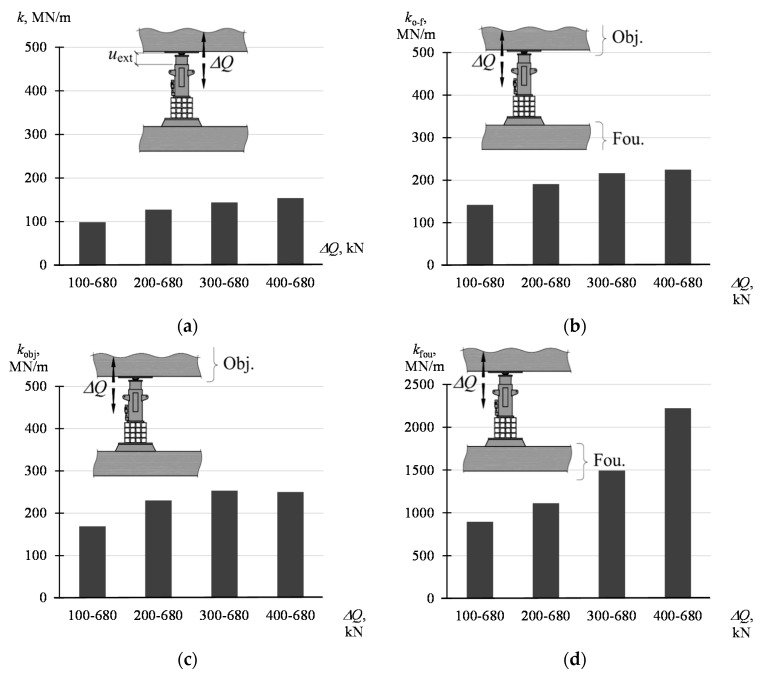
Stiffness of the system and its elements corresponding to the change Δ*Q* in the range of *Q*_min_–*Q*_max_: (**a**) *k* acc. to (4), (**b**) *k*_o-f_ acc. to (7), (**c**) *k*_obj_ acc. to (5), (**d**) *k*_fou_ acc. to (6).

**Figure 10 materials-13-02015-f010:**
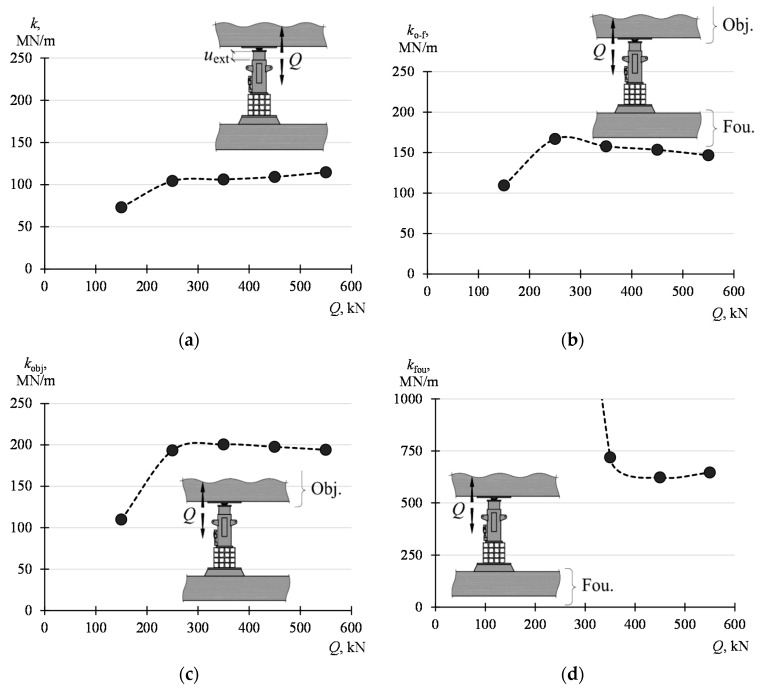
Influence of the value *Q* on the stiffnesses of the system and elements of the system: (**a**) *k*, (**b**) *k*_o-f_, (**c**) *k*_obj_, (**d**) *k*_fou._

**Figure 11 materials-13-02015-f011:**
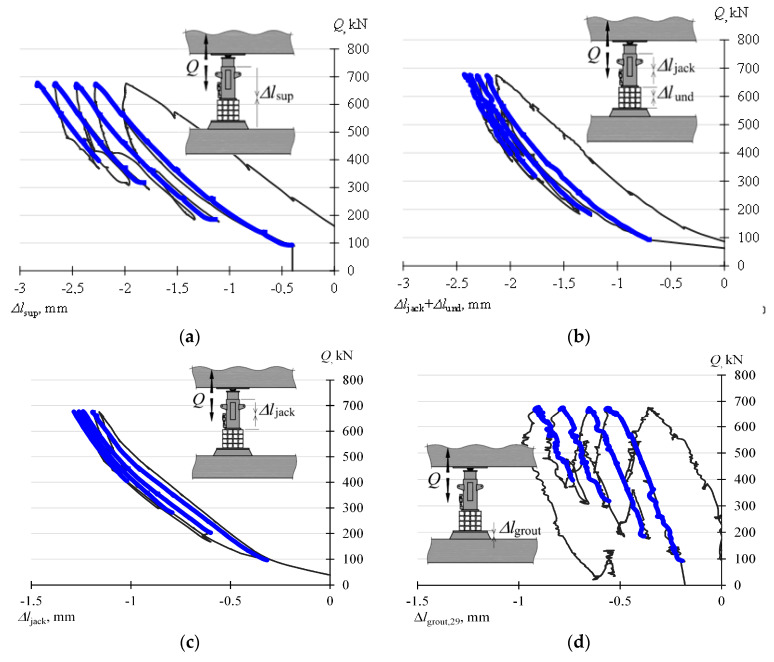
Changes in the length of the temporary support and its elements: (**a**) Δ*l*_sup_, (**b**) Δ*l*_jack_ + Δ*l*_und_, (**c**) Δ*l*_jack_, (**d**) Δ*l*_grout_.

**Figure 12 materials-13-02015-f012:**
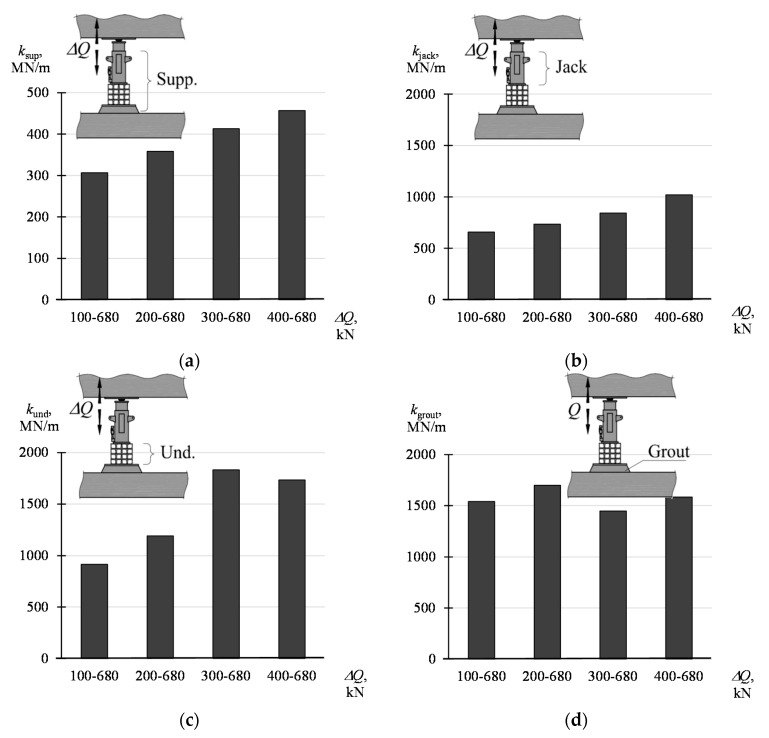
Stiffness of the support and its elements corresponding to the change Δ*Q* in the range of *Q*_min_–*Q*_max_: (**a**) *k*_sup_, (**b**) *k*_jack_, (**c**) *k*_und_, (**d**) *k*_grout._

**Figure 13 materials-13-02015-f013:**
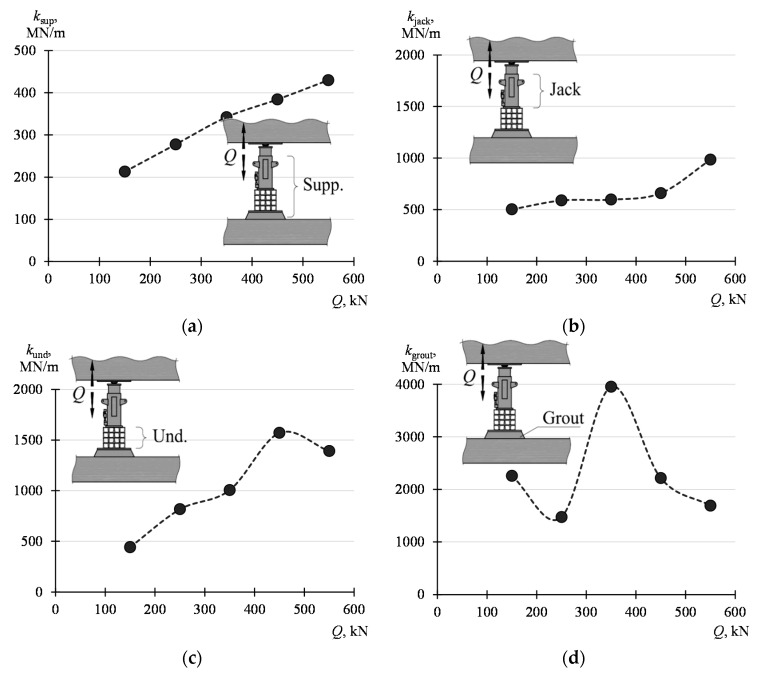
Influence of the value *Q* on the stiffnesses of the support and elements of the support: (**a**) *k*_sup_, (**b**) *k*_jack_, (**c**) *k*_und_, (**d**) *k*_grout._

**Figure 14 materials-13-02015-f014:**
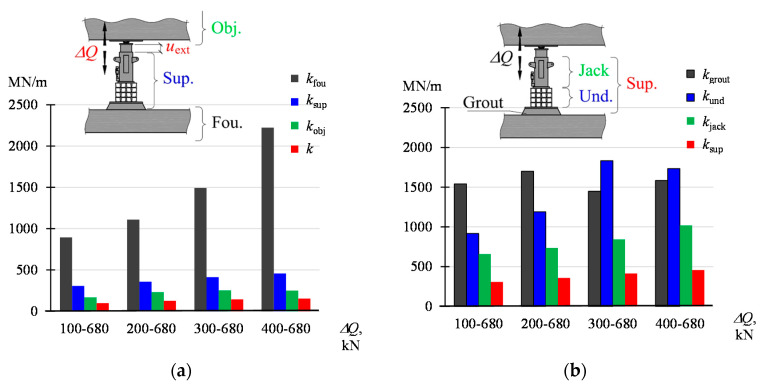
Stiffnesses corresponding to the change Δ*Q* in the range of *Q*_min_–*Q*_max_: (**a**) *k*_fou_, *k*_sup_, *k*_obj_, *k*, (**b**) *k*_grout_, *k*_und_, *k*_jack_, *k*_sup._

**Figure 15 materials-13-02015-f015:**
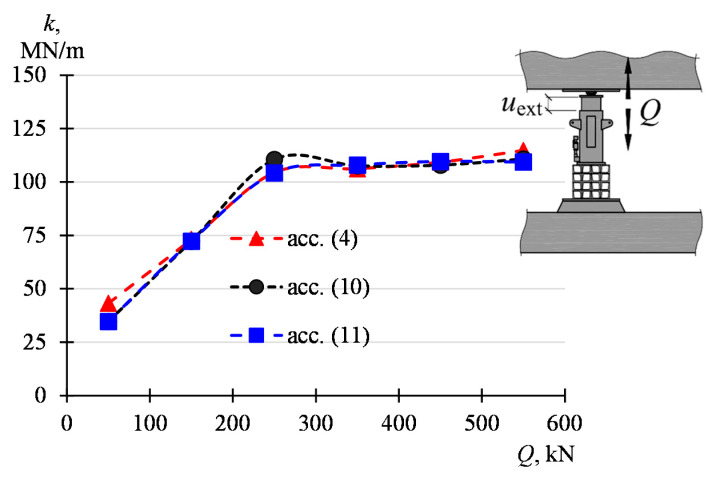
The stiffness *k* of the system determined based on (4), (10) and (11).

**Figure 16 materials-13-02015-f016:**
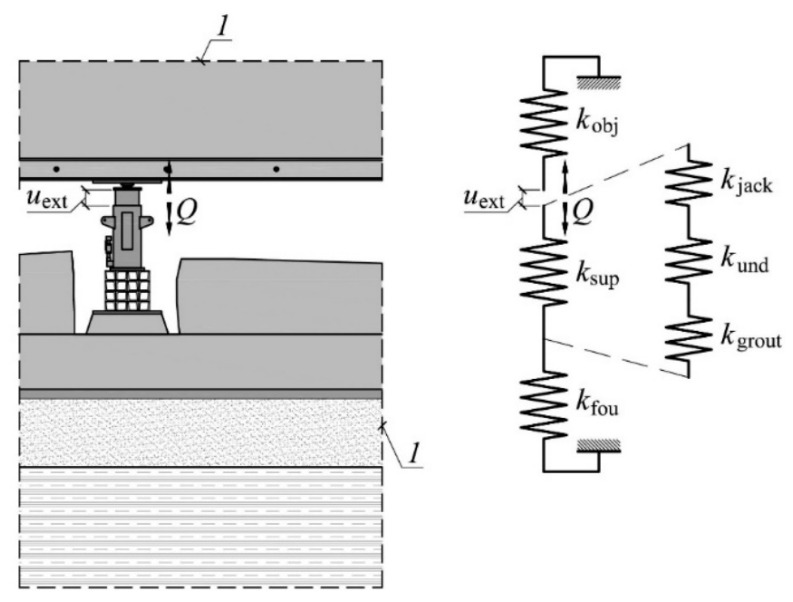
Dependencies between the analyzed stiffnesses; *1*—boundary conditions.

**Figure 17 materials-13-02015-f017:**
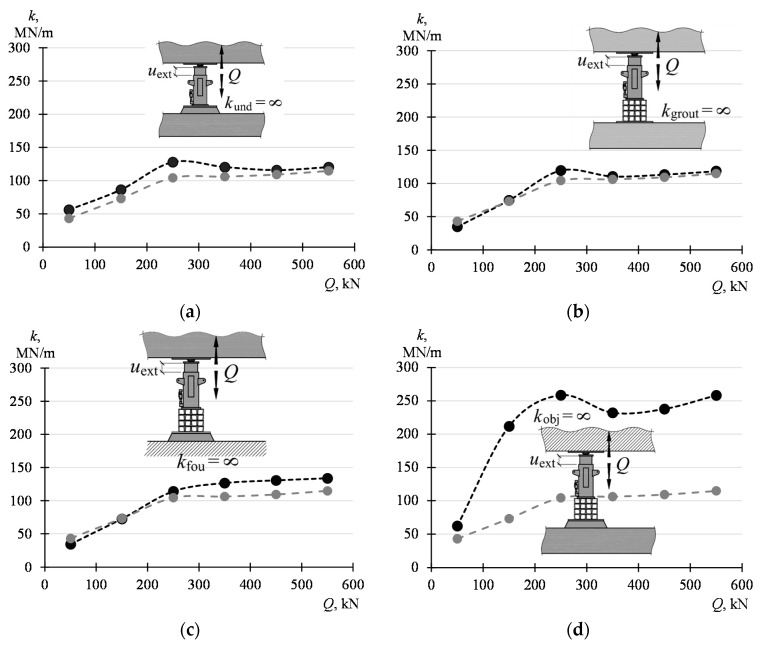
Analysis of the influence of stiffness of the components on the stiffness *k* of the system (the grey line indicates the respective values from Figure 15): (**a**) *k*_und_, (**b**) *k*_grout_, (**c**) *k*_fou_, (**d**) *k*_obj_.

**Figure 18 materials-13-02015-f018:**
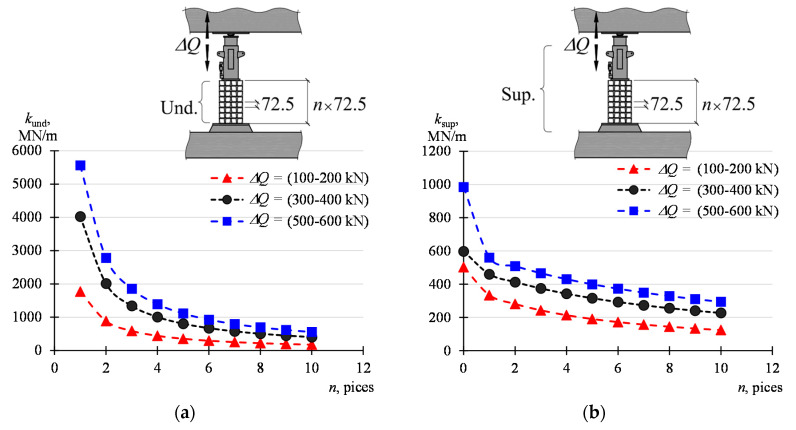
Influence of the number *n* of the parallelepiped elements on the stiffness of: (**a**) underlay, (**b**) support.

**Table 1 materials-13-02015-t001:** The values of extension, displacements and stiffness of the system and its elements corresponding to the change Δ*Q* in the range of *Q*_min_–*Q*_max_.

*Q*_min_–*Q*_max_	*u* _ext_	*u* _o-f_	*u* _obj_	*u* _fou_	*k*	*k* _o-f_	*k* _obj_	*k* _fou_
kN	mm				MN/m			
100–780	6.073	4.223	3.552	0.670	99	142	169	895
200–780	4.708	3.147	2.607	0.540	127	191	230	1111
300–780	4.157	2.770	2.368	0.402	144	217	253	1493
400–780	3.897	2.668	2.398	0.270	154	225	250	2223

**Table 2 materials-13-02015-t002:** Displacements of the building parts and stiffnesses of the system and elements of the building.

*Q*_min_–*Q*_max_	*u* _ext_	*u* _o-f_	*u* _obj_	*u* _fou_	*k*	*k* _o-f_	*k* _obj_	*k* _fou_
kN	mm				MN/m			
100–200	1.368	0.914	0.911	0.003	73	109	110	32452
200–300	0.958	0.599	0.517	0.027	104	167	193	3690
300–400	0.942	0.634	0.498	0.139	106	158	201	719
400–500	0.916	0.652	0.506	0.160	109	153	198	623
500–600	0.871	0.682	0.515	0.155	115	147	194	646

**Table 3 materials-13-02015-t003:** Changes in the length and stiffness of the support and its elements in the range of *Q*_min_–*Q*_max._

*Q*_min_–*Q*_max_	Δ*l*_sup_	Δ*l*_jack_ + Δ*l*_und_	Δ*l*_jack_	Δ*l*_und_	Δ*l*_grout_	*k* _sup_	*k* _jack-und_	*k* _jack_	*k* _und_	*k* _grout_
kN	mm					MN/m				
100–780	1.956	1.567	0.912	0.655	0.389	307	383	658	916	1541
200–780	1.395	1.101	0.681	0.420	0.294	358	454	734	1190	1700
300–780	0.969	0.692	0.474	0.218	0.276	413	578	844	1832	1448
400–780	0.657	0.467	0.294	0.173	0.189	457	642	1020	1734	1585

**Table 4 materials-13-02015-t004:** Changes in the length of the support and its elements corresponding to the change *Q* in the range of *Q*_min_–*Q*_max_.

*Q*_min_–*Q*_max_	Δ*l*_sup_	Δ*l*_jack_+Δ*l*_und_	Δ*l*_jack_	Δ*l*_und_	Δ*l*_grout_	*k* _sup_	*k* _jack_und_	*k* _jack_	*k* _und_	*k* _grout_
kN	mm					MN/m				
100–200	0.469	0.439	0.199	0.226	0.044	213	228	503	443	2258
200–300	0.360	0.310	0.170	0.122	0.068	278	323	590	816	1474
300–400	0.292	0.273	0.167	0.099	0.025	342	366	597	1006	3954
400–500	0.260	0.216	0.151	0.064	0.045	384	462	660	1571	2217
500–600	0.233	0.174	0.102	0.072	0.059	430	574	985	1391	1692

**Table 5 materials-13-02015-t005:** The stiffnesses of the system determined based on (4), (10) and (11).

*Q*_min_–*Q*_max_	*k* According to (4)	*k* According to (10)	*k* According to (11)
kN	MN/m	MN/m	MN/m
100–200	73	72	72
200–300	104	111	104
300–400	106	108	108
400–500	109	108	110
500–600	115	111	109

**Table 6 materials-13-02015-t006:** Stack stiffness depending on the number *n* of the parallelepiped elements forming the stack for different intervals Δ*Q*.

*Q*_min_–*Q*_max_	*n* = 1	*n* = 2	*n* = 3	*n* = 4	*n* = 5	*n* = 6	*n* = 7	*n* = 8	*n* = 9	*n* = 10
kN	MN/m									
100–200	1772	886	591	443	354	295	253	222	197	177
200–300	3264	1632	1088	816	653	544	466	408	363	326
300–400	4024	2012	1341	1006	805	671	575	503	447	402
400–500	6284	3142	2095	1571	1257	1047	898	786	698	628
500–600	5564	2782	1855	1391	1113	927	795	696	618	556

**Table 7 materials-13-02015-t007:** Support stiffness depending on the number *n* of the parallelepiped elements forming the stack for different intervals Δ*Q.*

*Q*_min_–*Q*_max_	*n* = 0	*n* = 1	*n* = 2	*n* = 3	*n* = 4	*n* = 5	*n* = 6	*n* = 7	*n* = 8	*n* = 9	*n* = 10
kN	MN/m										
100–200	503	334	281	242	213	190	172	157	144	133	124
200–300	590	373	335	304	278	256	237	221	207	195	184
300–400	597	460	413	374	342	316	293	273	255	240	227
400–500	660	471	438	409	384	362	342	325	309	294	281
500–600	985	560	509	466	430	399	372	349	328	310	294

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
