# Peer review of "In Situ Experimental Study on the Active Support Used for Building Rectification"

_materials, 2020, doi:10.3390/ma13092015_

Round 1
Reviewer 1 Report
This manuscript describes an experimental study on the use and effectiveness of the active support used for building rectification. The topic of the work is very interesting and relevant to the engineering community as more and more applications of building are being evaluated. The authors study focuses on the experimental study. While the topic is interesting, the reviewer has some general question regarding the technical content. The authors present a series of discussions surrounding the different measurement locations/types that needs to be revisited. The paper needs a lot of work to be considered as a suitable journal article.
- The literature review merely lists the published work and does not present the major findings from each effort and how it ties to the presented research. The introduction does not identify the knowledge gap that the authors are trying to address with their research.
- The objective statements are rather vague and lacks projected outcomes or how the paper will assist practitioners.
- Some unnecessary descriptions of the test results should be removed, and more in-depth analysis should be added. The quality of figures must be improved.
- Your conclusions are not well supported by your experimental data and analysis. Please clearly list the new, key findings supported by the experimental investigation. The conclusions should be modified after revision.
- The aspects of novelty of the paper must be pointed out more clearly due to the importance and complexity of the subject addressed in the work, particularly in the abstract and concluding remarks sections
- The authors should rewrite the abstract to include a summary of the key conclusions, clearly state the purpose of the work, the scope of the effort, the procedures used to execute the work, and major findings.
- The manuscript needs a thorough review for language and technical writing issues.
- Check references
- Arrange keywords in alphabetical order
- Resolution of some figures is low
Author Response
The author appreciates the insightful review and numerous comments to the manuscript. The clarity of the presented work will be enhanced and its quality improved by considering the comments.
The investigations designed and carried out concerned one issue, which was to determine the stiffness of a temporary support. The investigations were performed in in-situ conditions during a complex process of building rectification. The investigations themselves were therefore very limited in their scope. Hence, the analyses and conclusions refer only to the support stiffness.
With regard to the Reviewer's comments I note that I have made every effort to take them into account.
The literature review has been extended with the results of research of stacks of reinforced concrete and masonry elements. The extended section is at the end of Chapter 1. In turn, the quoted literature on the vertically deflected buildings was discussed in more detail at the beginning of this chapter. In addition, it was shown in the Chapter that no studies on temporary supports used to remove building leanings under in-situ conditions have been carried out so far.
The revised version of the article, in particular the summary, points to the practical application of the conclusions drawn from the undertaken studies.
The resolution of all figures in the article has been increased. This also concerns the diagrams and the images. In addition, small parts of the text that did not increase the transparency of the work were removed from the descriptions.
The conclusions in point 5 of the article have been entirely redrafted to reflect the results of the investigations and analyses carried out.
No investigations into the supports used to remove building deflections have been carried out to date, as indicated in the summary and at the end of Chapter 1. Likewise, the stiffness of the entire system, which includes the elevated part of the building, the part remaining in the ground and the supports, was not determined, as also noted in the conclusions.
The abstract has been completely redrafted so that it contains basic conclusions, defines the aim and scope of the investigations. Moreover, the keywords have been arranged in alphabetical order.
The valuable comments in the review will also help to design further research into the rectification of tilted buildings by their uneven lifting. This complex issue has not been the subject of research conducted so far.
Author
Reviewer 2 Report
In the paper entitled “In Situ Experimental Study on the Active Support used for Building Rectification” the experimental rectification of a five-floor vertically deflected building was carried out by installing 36 temporary supports in the cellar walls for this aim, the building was torn apart, and then unevenly lifted. The investigations carried out showed that it is advantageous to use supports with smaller stiffness for rectification. When using supports with smaller stiffness, it is necessary to induce forces with smaller values, which is beneficial. However, the use of supports with insufficient stiffness can be dangerous due to the loss of system stability. Moreover, in order to ensure that the unlifted part of the building does not deepen into the ground, it is necessary to change the values of forces in the supports by as little as possible, which means that small extensions have to be applied to the pistons of jacks. This is due to the fact that while the forces in the supports are growing, the stiffness of the part of the building remaining in the ground is decreasing.
---The paper is interesting and can be accepted for publication.
Comments:
The Abstract of the paper should be improved so that to better describe this paper and make it more interesting to the reader.
Author Response
The author appreciates the review comments to the manuscript.
The abstract has been completely redrafted so that it contains basic conclusions, defines the aim and scope of the investigationsIt was done to better describe the paper and make it more interesting to the reader.
Reviewer 3 Report
File attached

Author Response
The author appreciates the insightful review and comments to the manuscript. Thera are answers to the Reviewer's comments bellow.
The distance between the jacks was less than 2.63 m. The exact location of the jacks was due to the technical possibilities of making an opening for the jack (location of the central heating installation, electrical installation and others).
Measurements were carried out at the place where the vertical displacement was the largest. It was mentioned in the corrected version of the manuscript in line 118.
During rectification, the cylinder heads formed a plane. Consequently, the raised part of the building did not experienced significant deformations. After rectification, there were no scratches in the building walls.